# Comparison of Multiscale Imaging Methods for Brain Research

**DOI:** 10.3390/cells9061377

**Published:** 2020-06-01

**Authors:** Jessica Tröger, Christian Hoischen, Birgit Perner, Shamci Monajembashi, Aurélien Barbotin, Anna Löschberger, Christian Eggeling, Michael M. Kessels, Britta Qualmann, Peter Hemmerich

**Affiliations:** 1Institute of Biochemistry I, Jena University Hospital—Friedrich Schiller University Jena, Nonnenplan 2-4, 07743 Jena, Germany; Jessica.Troeger@med.uni-jena.de; 2Core Facility Imaging, Leibniz Institute on Aging – Fritz Lipmann Institute (FLI), Beutenbergstraße 11, 07745 Jena, Germany; christian.hoischen@leibniz-fli.de (C.H.); birgit.perner@leibniz-fli.de (B.P.); shamci.monajembashi@leibniz-fli.de (S.M.); 3Molecular Genetics Lab, Leibniz Institute on Aging—Fritz Lipmann Institute (FLI), Beutenbergstraße 11, 07745 Jena, Germany; 4Department of Engineering Science, University of Oxford, Parks Road, Oxford OX13PJ, UK; aurelien.barbotin@dtc.ox.ac.uk; 5Advanced Development Light Microscopy, Carl Zeiss Microscopy GmbH, Carl-Zeiss-Promenade 10, 07745 Jena, Germany; anna.loeschberger@zeiss.com; 6MRC Human Immunology Unit & Wolfson Imaging Center Oxford, MRC Weatherall Institute of Molecular Medicine, University of Oxford, Oxford OX39DS, UK; christian.eggeling@uni-jena.de; 7Dep. Biophysical Imaging, Leibniz Institute of Photonic Technology, Albert-Einstein-Straße 9, 07745 Jena, and Institute for Applied Optics and Biophysics, Faculty of Physics and Astronomy, Friedrich Schiller University Jena, Max-Wien-Platz 1, 07743 Jena, Germany

**Keywords:** advanced light microscopy, super-resolution, multi-scale imaging, tissue, brain

## Abstract

A major challenge in neuroscience is how to study structural alterations in the brain. Even small changes in synaptic composition could have severe outcomes for body functions. Many neuropathological diseases are attributable to disorganization of particular synaptic proteins. Yet, to detect and comprehensively describe and evaluate such often rather subtle deviations from the normal physiological status in a detailed and quantitative manner is very challenging. Here, we have compared side-by-side several commercially available light microscopes for their suitability in visualizing synaptic components in larger parts of the brain at low resolution, at extended resolution as well as at super-resolution. Microscopic technologies included stereo, widefield, deconvolution, confocal, and super-resolution set-ups. We also analyzed the impact of adaptive optics, a motorized objective correction collar and CUDA graphics card technology on imaging quality and acquisition speed. Our observations evaluate a basic set of techniques, which allow for multi-color brain imaging from centimeter to nanometer scales. The comparative multi-modal strategy we established can be used as a guide for researchers to select the most appropriate light microscopy method in addressing specific questions in brain research, and we also give insights into recent developments such as optical aberration corrections.

## 1. Introduction

Neurological disorders and also aging have significant effects on different brain functions. Factors determining the levels of cognitive impairments seem to be related to biochemical, cell biological and physiological processes in the brain, which result in altered neurotransmission processes and patterns as well as in reduced neurotrophic support of neural cells but also in metabolic, hormonal and immune dysregulation, increased oxidative stress and inflammation. These deficits have a strong impact on brain performance and life quality. Considering that the brain maintains a precise, yet dynamic, balance between excitation and inhibition, it is not surprising to see that neurological dysfunctions often affect both excitatory and inhibitory neurotransmission. Excitatory neurons are a large majority in the brain, yet local inhibitory interneurons modulate their firing and timing. Thus, it is well conceivable that disruptions of both excitatory and inhibitory circuits are responsible for clinical features of many neuropsychiatric and neurodegenerative disorders, such as schizophrenia, autism, depression, epilepsy, Alzheimer’s disease and Parkinson’s disease, but also of age-related deficits, as their careful balancing and crosstalk are no longer ensured [1,2,3,4,5,6,7,8,9].

Besides proper balance between excitation and inhibition in different brain regions and neuronal circuits, proper brain function also depends on a delicate balance between firmly established neuronal circuits and synaptic contacts. Maintenance, repair and plasticity processes occur at the level of dendritic arborization and axonal projection, as well as at the level of the synaptic contacts between neuronal cells. Synapses play a central role in cognitive functions [10]. A decline in neuron numbers, a reduction of soma size, a pruning of the dendritic arbor, reductions of synaptic contacts, loss of dendritic spines and structural changes within both pre- and the postsynaptic morphology and organization, such as changes in neurotransmitter receptor organization, have been detected as linked to neuropathophysiological and age-related structural changes across the nervous system [8,11]. Interestingly, these declines seem in part to be interlinked, as e.g., synaptic dysfunction and pruning of synaptic contacts may precede neuronal loss [1,5,6,7,12].

The communicative interface between the presynapse with its neurotransmitter-filled synaptic vesicles and the juxtaposed postsynaptic compartment were found to take center stage in neurological diseases as well as in aging [11,13,14,15]. At the presynaptic side, this communicative interface contains a signal-responsive array of proteins at the so-called active zone that effectively and precisely steers the presynaptic neurotransmitter release. The postsynaptic side represents a specialized compartment of signal reception and integration and is composed of a dense protein array —the postsynaptic density (PSD). The PSD mainly contains neurotransmitter receptors, signaling components and scaffold proteins, which among others include Homer and ProSAP/Shanks. In the case of excitatory glutamatergic synapses, this postsynaptic compartment is often located in dendritic spines protruding maximally 1–3 µm from the dendrite [13,16,17].

It is generally accepted that structural organization in synapses is linked to synaptic function. This holds true for both presynapses, where the organization of the active zone and the organization of the different pools of synaptic vesicles have consequences for the efficacy of neurotransmitter release, and for postsynapses, where the organization of receptors as well as of down-stream signaling and scaffold components are strongly linked to synaptic performance and plasticity. Yet, the pathophysiology of neurological diseases and of aging in the brain seems to be quite complex. Microscopic observations of neuronal morphology seem to differ when different species, brain regions, cell types and sexes are compared by different labs [4,6,8,14]. Furthermore, pathophysiological structural defects in the brain often are relatively subtle and it remains somewhat unclear how much of this lack of detection of structural differences in some studies and/or brain regions is merely due to a lack of lateral and axial resolution, as the structures involved are near or below the optical resolution limit of light microscopy.

Dendritic spines harboring postsynapses are maximally 1–3 µm in length and their neck is e.g., only about 100 nm wide. Presynapses and postsynapses in most cases only have a size of 0.2–1 µm diameter. Synaptic vesicles are about 40 nm in diameter. Synaptic substructures, such as the active zone or the PSD, both usually only show maximal extensions of ~350 nm [18,19]. Imaging of individual cells and cellular compartment within brain tissue samples also can only be done if closely adjacent structures from distinct cells can clearly be distinguished and resolved. This in average also requires resolutions below 100 nm. Such resolutions can easily be reached by electron microscopy (EM) of ultrathin cut brain sections (practical resolution limit, 1–5 nm). Therefore, a significant portion of our knowledge of synaptic organization as well as of pathophysiological defects is based on EM analyses [20].

In contemporary brain research, however, one challenge is to image and quantitatively examine larger parts of the brain in high resolution and to do this in 3D. It is technically very challenging to accomplish this by EM and instead rather brings light microscopy-based imaging on into play. Over the last decades several microscopy techniques have been implemented in biological and medical research which allow multi-modal analysis of whole organs from the millimeter to the nanometer range, both in fixed as well as in living specimen [21]. In particular optical super-resolution microscopy has been intensively employed to examine structural aspects of synapses and contributed fascinating new findings in neuroscience and synapse organization [22,23,24]. Highly advanced and specialized microscope platforms have been developed in some imaging labs world-wide [25,26]. Currently, however, biologists are unable to benefit from many of these developments because they have outpaced commercialization [27,28].

We therefore set out to assess and compare the performance of nine different commercially available light microscopy techniques, including stereo, widefield, widefield-deconvolution, point scanning conventional and sub-Airy pinhole-sized as well as spinning disc confocal, Airyscan, structured illumination (SIM) and stimulated emission depletion (STED) technologies, to establish a basic set of multi-scale protocols, which allow for multi-color brain imaging from centimeter to nanometer scales. We used one and the same brain tissue specimen on all microscopic platforms. This enabled us to directly compare the performance of different microscope techniques with respect to image quality and acquisition time. We also tested additional modalities to improve light microscopy and data processing by including compute unified device architecture (CUDA) graphics cards, motorized correction collar objectives and adaptive optics (AO) in our tests. The data from our comparative efforts will be helpful as a guide to select the most appropriate commercially available equipment, for example in an institutional microscope facility, to address specific biological questions, not only in brain research.

## 2. Materials and Methods

### 2.1. Antibodies and Reagents

Guinea pig polyclonal anti-Homer (anti-Homer1; #160004) and polyclonal rabbit anti-Shank2/ProSAP1 (#162202) antiserum were obtained from Synaptic Systems (Göttingen, Germany) and mouse monoclonal anti-MAP2 (clone HM-2; M4403) was from Sigma Aldrich (St. Louis, MO, USA). As secondary antibodies STAR635P-labelled goat anti-guinea pig IgGs or STAR580-labelled goat anti-rabbit IgGs (# 2-0112-007-1 and # 2-0012-005-8, respectively; Abberior, Göttingen, Germany) were applied. Additionally, Alexa Fluor^®^488-labeled donkey anti-mouse antibody (R37114; ThermoFisher Scientific, Waltham, MA, USA) was used.

### 2.2. Immunolabeling of Mouse Brain Sections

Adult male mice were sacrificed by overdose CO_2_ inhalation and perfused with PBS and 4% paraformaldehyde (PFA) in PBS in a transcardial manner as described previously [29]. Brains were post-fixed overnight in 4% (*w*/*v*) PFA at 4 °C and subsequently transferred into 10% (*w*/*v*) and 30% (*w*/*v*) sucrose solution for 24 h each. Sagittal brain sections were sliced with 60 µm thickness on a sliding microtome (SM 2000R; Leica, Wetzlar, Germany) and collected in anti-freezing solution (15% glucose *w*/*v*, 30% ethylenglycol *v*/*v*, 0.02% sodium azide *w*/*v*, 50% phosphate buffer (PB) (PB, 77.4 mM Na_2_HPO_4_ and 22.6 mM NaH_2_PO_4_ yielding pH 7.4) as free-floating slices. For immunolabeling, brain sections were washed with PB, permeabilized and blocked with 5% (*v*/*v*) goat serum, 0.25% (*v*/*v*) Triton X-100 in PB (blocking solution) for 1 h at RT. Primary and secondary antibody incubations were done in blocking solution for 48 h at 4 °C each. After washing and DAPI staining (30 min, 1:1000 in PB), slices were mounted onto glass slides (Menzel, Braunschweig, Germany) with Fluoromount-G (Southern Biotech, Birmingham, AL, USA).

### 2.3. Culturing, Transfection and Immunostaining of Primary Rat Hippocampal neurons

Primary rat hippocampal neuronal cells for immunofluorescence analyzes were cultured and maintained as described previously [30,31,32]. Briefly, neurons dissociated from hippocampi of E18 rats were seeded at a density of 60,000/well (24-well plate). Cells were maintained in Neurobasal™ medium containing 2 mM l-glutamine, 1× B27 and 1 µM/ml penicillin/streptomycin and were kept at 37 °C with 90% humidity and 5% CO_2_. Neurons were fixed at DIV16 in 4% (*w*/*v*) PFA in PBS pH 7.4 at RT for 3–6 min and permeabilized in block solution containing 10% (*v*/*v*) horse serum, 5% (*w*/*v*) BSA in PBS with 0.2% (*v*/*v*) Triton^®^ X-100. Antibody incubations were performed in the same buffer without Triton^®^ X-100 according to Kessels et al. [33] and Pinyol et al. [34]. Briefly, neurons were incubated with primary antibodies overnight at 4 °C, washed three times with block solution and then incubated with secondary antibodies (1 h, RT). After final washing steps and DAPI staining (5 min, 1:10,000 in PBS) coverslips were mounted onto glass slides using Mowiol.

### 2.4. Confocal Microscopy

Confocal test images were recorded using either a TCS SP5 microscope (Leica; equipped with 40×/0.75dry and 63×/1.4 oil objectives, hybrid detectors and LAS AF software).

Image processing including conversion of imaged z-stacks into maximum intensity projections (MIP) was done in ImageJ [35] or Adobe Photoshop software (Adobe, Mountain View, CA, USA).

### 2.5. STED Microscopy 

Gated 2D-STED images were acquired on a Leica TCS SP8 STED microscope equipped with a 100×/1.4 N.A. objective according to [36]. Pixel size in STED acquisition was applied automatically in LAS-X software (Leica) for the most red-shifted dye (STAR 635P), usually resulting in a pixel size of less than 20 × 20 nm. STED beam alignment was performed between the pulsed white light laser and 592 nm depletion laser before each imaging session. DAPI, Alexa Fluor 488, STAR 580 and STAR 635P were excited with laser lines 405 nm, 488 nm, 580 nm and 635 nm of the white light laser, respectively. Emission was captured through band pass settings 430–470 nm, 505–550 nm, 590–620 nm and 648–720 nm, respectively. Depletion of STAR 580 and STAR 635P was performed with the 775 nm depletion laser. Alexa Fluor 488 was depleted with the 595 depletion laser. The 775 nm depletion laser (model PFL-2000-775-B1R, MPB Communications Inc., Point-Claire, Quebec, Canada) was pulsed (repetition rate 80 MHz; pulse length 0.71 ns). We used the vortex phase mask only to maximize lateral resolution. The power of the depletion laser was optimized for each dye to obtain highest resolution while avoiding bleaching. Imaging conditions were fine-tuned on several regions before application of the optimized settings for final stacks. Each dye was imaged in sequential scans to avoid spectral mixing. Hybrid detector gain was fixed to 100%, while excitation wavelength intensity was set such to prevent pixel saturation. Images were obtained using a pixel dwell time of 100 ns. Photon time gating was employed by collecting lifetimes between 0.3 and 6.0 ns for the STAR dyes and 1.5–3.0 for Alexa Fluor 488. To compensate for inevitable signal intensity loss during STED acquisition, the excitation laser power was set 3- to 5-fold higher than in conventional confocal mode. In STED channels, the pinhole was set 1 Airy Units. In the non-STED channels (DAPI) the pinhole was set to 0.49 Airy Units to allow for super-resolution confocal microscopy according to the HyVolution mode of the Leica SP8 microscope. All images were deconvolved with Huygens Professional Software (Scientific Volume Imaging B.V., Hilversum, The Netherlands) using the specific deconvolution pre-settings in Huygens software dedicated for STED on the SP8 optics.

### 2.6. Adaptive Optics (AO) z-STED Microscopy

AO z-STED microscopy was performed on a modified STED/RESOLFT microscope (Abberior, Göttingen, Germany) described in [37]. Excitation was performed by a diode laser pulsed at a frequency of 80 MHz and a wavelength of 640 nm. Depletion was realized with a z-STED depletion pattern (the so-called “bottle beam”). The STED laser was pulsed at a frequency of 80 MHz at a wavelength of 755 nm and an average power of 82 mW measured in the back focal plane of the objective. The original pulse is approximately 100 fs long according to the manufacturer, and we stretched it with a 40 cm glass rod and a 100 m single-mode fiber. AO was performed by first introducing spherical aberrations to compensate the effects of the refractive index mismatch between the immersion medium of the objective and that of the brain tissues, as described for instance in [38]. Remaining aberrations were corrected using the sensorless method as described in [39], using instead a wavelet-based image quality metric to assess image quality [40].

### 2.7. Widefield Fluorescence and Structured Illumination Optical Sectioning (SIOS) Microscopy

Widefield fluorescence microscopy was performed on an AxioOberver microscope (Zeiss) equipped with a HXP lamp or LED illumination (Colibri) (Zeiss), an AxioCam 702 mono camera (Zeiss), and a Plan-Apochromat 20×/0.8 objective (Zeiss). Images and image stacks were acquired in ZEN 2.3 software (Zeiss). Quality control experiments using cultured neurons were done using a 63×/1.4 N.A. objective. Where indicated, an ApoTome.2 grid (Zeiss) was inserted into the optical path to produce SIOS images. In ApoTome imaging, an optical grid is tilted back and forth in the light path and projected onto the specimen. At least three raw images of the specimen were acquired at different positions of the grid. The software used the grid-projected images to calculate and remove the out-of-focus light and combined the 3 images into 1. In case of HXP lamp illumination, the light intensity was set to 100% to minimize exposure times. In experiments with LED illumination the light intensity was varied along with the exposure time to minimize photobleaching of the tissue sample.

### 2.8. Spinning Disc Confocal Microscopy (SPDM)

For spinning disc confocal microscopy (SPDM) we used a Zeiss Observer Z1 inverted microscope (Zeiss) equipped with a 63×/1.4 N.A. C-Apochromat objective, a spinning disc confocal scanning system CSU-X1 (Yokagawa Electric Corporation, Tokyo, Japan), and 405 nm, 488 nm, 568 nm and 633 nm diode lasers. Images were captured on an AxioCam MRm camera (Zeiss). Fluorescence of the four fluorophores in tissue sections (DAPI, Alexa Fluor 488, STAR 580, STAR 635P) were acquired in sequential scans employing ZEN blue software (Zeiss). Laser intensities and camera exposure times were adjusted to minimize photobleaching and overall scan time. Image processing was done using ZEN 2012 (Zeiss), and Adobe Photoshop software.

### 2.9. Airyscan Microscopy

Airyscan images were acquired in the SR (super resolution) mode on a Zeiss LSM 880 microscope equipped with an Airyscan detector using a Plan-Apochromat 63×/1.4N.A. oil DIC M27 objective. DAPI was excited with a 405 nm Diode at 0.2% laser power and the emission was detected through the following settings: single beam splitter (SBS) band path (BP) 420–460 + long path (LP) 500 in combination with emission filter BP 420–480 + BP 495–620 (detector gain 804). Alexa Fluor 488 was excited with a 488 nm argon laser line (0.12%) using main beam splitter (MBS) 488/561/633 and the emission was captured through an SBS LP 460 in combination with emission filter BP 420–480 + BP 495–550 (detector gain 931). STAR 580 was excited at 561 nm at 1.2% laser power using MBS 488/561/633 and the emission was captured through an SBS SP 615 in combination with emission filter BP 420–480 + BP 495–620 (detector gain 953). STAR 635P was excited at 633 nm at 1.0% laser power using MBS 488/561/633 and the emission was captured through an SBS LP 660 in combination with emission filter BP 570–620 + LP 645 (detector gain 931). Pixel size in Airyscan high resolution acquisition was applied automatically in ZEN 2.3 software (Zeiss) for the Alexa Fluor 488 Channel, usually resulting in a pixel size of 40 nm. The Pixel Dwell was 1.4 µsec. Z-Stack scans were performed at 0.16 µm intervals fulfilling Nyquist criteria. Tracks were changed after each Z-stack to reduce the number of SBS and time for emission filter changes. Images were deconvolved with the deconvolution tool of the ZEN Blue software (Zeiss).

### 2.10. HyVolution Imaging

HyVolution is based on a combination of confocal imaging using sub-Airy pinhole sizes (0.45–0.6 Airy Units, depending on the fluorophores used) with subsequent computational image deconvolution [41] as initially described by Schrader et al., [42] and Lam et al., [43]. HyVolution imaging was performed on a Leica TCS SP8 X-White Light Laser confocal microscope equipped with an inverted microscope (DMI 8; Leica), a 100× objective (HC PL APO CS2 100×1.4 oil), Hybrid detectors and HyVolution 2 software. Images were acquired as suggested by the HyVolution module within the LAS microscope software (Leica). Pre-settings in the HyVolution mode were established to achieve the highest possible optical resolution. HyVolution images of immunofluorescently labeled mouse tissue were acquired using the 405 nm, 488 nm, 580 nm and 635 nm laser lines of the white-light laser to excite DAPI, Alexa Fluor 488, STAR 580 and STAR 635P dyes, respectively. Fluorescence emission was recorded on two hybrid detectors (HyD) at 420–460 nm, 503–547 nm, 585–620 nm, and 645–719 nm, respectively. DAPI/STAR 580 and Alexa Fluor 488/STAR 635P channels were each captured simultaneously. The two channel pairs were scanned sequentially to avoid bleed-through and/or cross-talk between spectral channels. The HyVolution mode was set to ‘best resolution’, which included proper Nyquist adaption and 4-fold averaging.

The super-resolved confocal stacks were deconvolved in Huygens deconvolution professional suite software (Scientific Volume Imaging, Hilversum, The Netherlands) with GPU acceleration.

### 2.11. Stereo Microscopy

A Zeiss Axio Zoom.V16 microscope equipped with a 2.3× objective (Plan-NEOFLUAR 2.3×/0.57 FWD 10.6 mm) objective and an ApoTome.2 slider for optical sectioning was used.

### 2.12. Slide Scanner Microscopy

The whole brain tissue slice was imaged on a multi-slide scanning microscope (Axio Scan.Z1, Zeiss) equipped with a 20× objective (Plan-Apochromat, 20×/0.8), LED light source (Colibri) and an AxioCam 702 mono camera (Zeiss).

### 2.13. Lattice-SIM Microscopy

Lattice-SIM images were acquired on a Zeiss Elyra 7 AxioObserver microscope equipped with an Plan-Apochromat 63×/1.4 Oil DIC M27 objective and two pco.edge sCMOS (version 4.2 CL HS) cameras. The system contained 405 nm and 642 nm diode, and 488 nm and 561 nm OPSL lasers. For each focal plane 13 phase images were acquired. Exposure time and laser power were balanced for each fluorescence channel individually to minimize bleaching and exposure time. SIM and Lattice-SIM reconstruction was performed with the SIM processing Tool of the ZEN 3.0 SR (black) software.

Information on the specific hardware configurations and acquisition settings of all microscopy platforms are available on request.

### 2.14. Deconvolution

Confocal images stacks produced on Leica microscopes (Figures 3C and 5C, 6 HyVolution and 6 STED) were deconvolved with the Huygens 18.04 software (Scientific Volume Imaging, B.V., Hilversum, The Netherlands) using a theoretical point spread function (PSF) and the Classical Maximum Likelihood Estimation (CMLE) algorithm with 40 iterations. All parameters for deconvolution were automatically selected in Huygens software considering the metadata provided with the Leica image files. We also varied parameters such as signal to noise ratio (SNR) and background but did not find substantial improvements compared to the default settings in Huygens. Image stacks acquired on Zeiss microscopes (Figure 2C2d–6d, Figure 3A1–A5, Figure 3B4,B5, Figure 4A,B, Figure 5A1–A6 and Figure 6 SIOS, SPDM and Airyscan) were deconvolved in ZEN software.

We tested a variety of pre-settings, including Nearest Neighbor, Regularized Inverse Filter, Fast Iterative (FI) and Constrained Iterative algorithms. The FI algorithm resulted in high quality images with moderately short processing time scales. FI was therefore selected throughout in the adjustable settings mode in ZEN with the following values: Likelihood-Poisson-Meinel; regularization-zero order; optimization-numerical gradient; max. iterations-5. Deconvolution in Huygens or Zeiss ZEN software was accelerated using a NVIDIA Quadro P6000 CUDA graphics card.

## 3. Results

Brain research at the cellular level relies on the detailed imaging of neuronal structures in brain tissue slices, slice cultures and/or living animals. With the demand to image subtle changes in synaptic architecture, particular imaging strategies need to match the requirements in optical resolution. On the other hand, times of image acquisition and processing need to be reduced. We therefore applied different types of 3D multi-scale microscopy platforms in a comparative manner to one and the same tissue specimen. The aim was to dissect the optimal imaging modalities appropriate for addressing moderate changes in neuronal organization. Therefore, we focused on two components of the postsynapse, the postsynaptic scaffold proteins Homer and ProSAP1/Shank2. The antibodies against synaptic markers Homer and Shank2/ProSAP1 used for tissue immuno-fluorescence labeling were firstly quality-controlled on cultured rat hippocampal neurons recorded on a fluorescence widefield microscope using a 63×/1.4 N.A. objective (Figure 1A). As expected, Homer colocalized strongly with Shank2/ProSAP1 in a dot-like pattern along dendrites identified by microtubule-associated protein 2 (MAP2) immunolabeling as depicted in the enlarged image (Figure 1B).

We then proceeded to use this antibody combination in brain tissue section immunolabeling. One and the same tissue specimen was used on all microscopy platforms for direct comparison. The number of z-planes and pixel size for each microscopy technique was selected according to the Nyquist criterion, i.e., a factor of two smaller than the respective resolution estimated from the employed wavelength and numerical aperture [44]. The imaging software of all commercially available microscopes used in our study included automatic optimization of the imaging parameters according to Nyquist. This option was selected in all our analyses.

### 3.1. Tissue Imaging with Widefield Microscopy

In order to assess imaging in large-field microscopy, the performance of stereo, multi-slide scanning and conventional widefield fluorescence microscopy were compared (Figure 2). Working principles and technical details of stereo and widefield fluorescence microscope systems have been describe in detail previously [45,46,47]. First, we imaged the brain tissue on the stereo microscope, which was equipped with a 2.3× objective. On these images we could reveal the position of single cell nuclei identified by DAPI staining (Figure 2A3, arrow) but failed to detect the filamentous distribution of MAP2-labeled dendrites in the brain tissue (Figure 2A4). The anti-Shank2 immunofluorescence appeared as a quite strong and homogeneous signal throughout the specimen with no local spots of preferred staining indicative of individual synapses (Figure 2A5). Since the antibody did not produce any unspecific staining of cultured hippocampal neurons (Figure 1), the strong staining in brain tissue likely represents background fluorescence within which the stereo microscope fails to detect single synapses.

The very same tissue slice was then 3D-imaged using a slide scanner employing a 20× objective (Figure 2B1–B6). The slide scanner revealed chromocenters in the DAPI-stained cell nuclei, which presumably represented accumulations of centromeric heterochromatin (Figure 2B3, arrows) [48]. MAP2-immunopositive dendrites were also detected (Figure 2B4, arrow). As expected, the superior lateral resolution of the slide scanners’ 20× objective compared to the stereo microscope delivered less background of Shank2 or Homer fluorescence, however, single synapses were still not revealed (Figure 2B5,B6).

Essentially the same observations were made using a conventional widefield fluorescence microscope equipped with similar optics compared to the slide scanner (Figure 2C1–C6). We conclude that, at least under the conditions and specimen applied here, widefield microscopy is suitable to detect larger subcellular structures in the 60 µm thick brain tissue section analyzed here but is less appropriate to image very small structures, such as synapses.

We also comparatively assessed the time required for image acquisition and processing of the complete multi-fluorescence mouse brain tissue slice (volume: 4 × 10^9^ µm^3^) in 3D (Table 1). Here, the stereo microscope (40 min acquisition time) clearly outperformed the other two widefield systems (slide scanner: 75 min, widefield: > 10 h) (Table 1).

In an attempt to further improve image quality and/or optical resolution we subjected a sub-volume of the 3D widefield data set to deconvolution. Deconvolution is a computational method used to reduce out-of-focus light in 3D microscope data sets [49]. Consideration of the same optical section before and after deconvolution revealed a tremendous improvement in SNR for the DAPI and the MAP2 channels (Figure 2C2d,C3d,C4d). Although deconvolution resulted in a punctate pattern of Shank2 or Homer distribution (Figure 2C5d,C6d), it was difficult to judge whether these structures indeed represented synapses, as there was almost no colocalization between Homer and Shank2 immunosignals (Figure 2C2d).

In summary, the widefield approaches tested here on the same tissue slice allow for rapid multi-color 3D scanning of large volumes with very low (stereo microscope) to moderate (slide scanner) resolution. Conventional widefield microscopy with a 20× objective delivered improved in optical resolution of large subcellular structures (here: chromocenters, dendrites) after deconvolution. However, acquisition/processing time lasted several days and file size increased to 840 GB (Table 1). If stitching and deconvolution would be optimized and implemented to occur simultaneously to image acquisition, the total time to collect the four fluorescence color channels in the 4 × 10^9^ µm^3^ tissue region used here could probably be reduced to an over-night scan.

### 3.2. Confocal 3D Imaging of Brain Tissue

As seen in the previous section, out-of-focus light adds a blurry background in thick specimen, whereby weak but specific signals may go undetected. Confocal laser point-scanning (CLSM) and SPDM have become state-of-the-art techniques to remove out-of-focus fluorescence, improve SNR and increase resolution [50,51]. The principles of CLSM and SPDM will not be presented here - we refer to excellent contributions on this topic for further reading [47,50,51,52,53]. A different approach to obtain optically sectioned images was realized by implementation of ‘structured light’ into a conventional fluorescence widefield microscope [54]. Here we refer to this approach as structured illumination optical sectioning (SIOS). In SIOS, a grid pattern is inserted into the excitation light beam between the light source and the sample. The grid projects a light pattern into the image plane. At least three images of the fluorescent sample are acquired at different grid positions and a simple image algorithm then removes the out-of-focus light in the final image. The (dis)advantages of this technique have been reported previously [50]. We would like to point out that SIOS as described and applied here is different from super resolution SIM (SR-SIM) [55,56], which is addressed in Section 3.3.

SIOS, SPDM and CLSM were performed on adjacent 90 × 90 × 50 µm^3^ volumes of the same specimen shown in Figure 2 using a 63× oil immersion objective. This approach allowed for direct performance comparison among the three different set-ups. SIOS results are shown in Figure 3A. Figure 3A1 shows a 3D representation of the 4-color image stack. A single optical section after SIOS image processing revealed dot-like foci containing Homer and Shank2 at MAP2-positive dendrites (Figure 3A2,A3). Foci size (well below 1 µm) and colocalization of Homer with Shank2 in the foci and their localization in close proximity to the dendritic arbor suggested that single synapses were visualized by SIOS in thick tissue (Figure 3A3, arrow). In the ZEN software (Zeiss), SIOS stacks can be alternatively processed by deconvolution which is shown in Figure 3A4,A5). Although at low resolution but consistent with the SIOS-processed image (Figure 3A3), deconvolution of the widefield data also revealed single synapses (Figure 3A5, arrow). In summary, 3D widefield microscopy of brain tissue using a 63× objective is sufficient to detect and resolve small subcellular structures (such as synapses) when quasi-confocal (SIOS) or deconvolution methods are applied. Acquisition time of the SIOS stack was 18 min 25 s (Table 1). SIOS processing lasted only 2.5 min while deconvolution took more than 8 min (Table 1). However, image quality after deconvolution appeared somewhat superior with respect to Homer and Shank2 fluorescence (compare Figure 3A3,A5).

Next, we acquired a SPDM image stack of a 90 × 90 × 50 µm^3^ volume from the same specimen (Figure 3B1). Although confocal, a single optical section of the image stack still displayed a high degree of haze for Homer and Shank2 staining while nuclei (including chromocenters) and MAP2-positive dendrites were clearly visible (Figure 3B2,B3). Deconvolution of the confocal image stack resulted in a considerable increase of the SNR of all fluorescence channels, which allowed unambiguous identification of single synapses (Figure 3B4,B5). Strikingly, acquisition and deconvolution time summed up to only 11 min, which, at comparable image quality, is only about half the time of SIOS imaging of the same stack (Table 1). This is mainly attributable to the fact that SIOS requires three (grid phase) images per optical section, while only one snapshot is necessary in SPDM.

Finally, the same tissue was imaged with an advanced point-scanning CLSM. We employed a combination of confocal imaging at sub-Airy pinhole size (0.45 - 0.6 Airy Units) with subsequent computational image deconvolution, as described more than two decades ago [42]. This approach is commercially realized as HyVolution mode on Leica confocal systems but sub-Airy imaging followed by third party deconvolution can also be performed on any other confocal system [43]. Another advantage of this hardware/software combination is a significant increase in optical resolution down to below 150 nm (*x*/*y*) and 200 nm in z direction [43].

HyVolution was employed here because it has the time-saving advantage of online deconvolution during image acquisition. The sub-Airy pinhole CLSM image stack is shown in Figure 3C1. A single confocal section from this stack (Figure 3C2) showed considerably less haze compared to its spinning disc counterpart (Figure 3B2). Deconvolution further improved SNR and image quality (Figure 3C3,C4) as well as optical resolution: single synapses with a diameter of about 200 nm now were clearly visible (Figure 3C5). The achieved high image quality and improved resolution, however, came at a price. The acquisition time with the confocal point scanner for the single confocal stack was 3 h 42 min (Table 1). This is not only attributable to the slowness of the point scanning device compared to camera-based systems but also to the increased number of optical sections (>400 vs. 200) necessary for improving axial resolution (Table 1).

We also assessed the tissue imaging penetration depth of SIOS compared to SPDM (Figure 4). In orthogonal sections the immunofluorescence signals of Homer and Shank2 were blurred in SIOS (Figure 4A) while single synapses were resolved by SPDM followed by deconvolution (Figure 4B). In addition, SPDM imaging delivered details throughout the entire z-stack while SIOS was substantially limited in z imaging direction, consistent with previously reported limitations of the optical sectioning approach [50]. We cannot confirm other reported limitations of SIOS, including noisy or mottled appearance or slow imaging speed [50], which is probably due to technological advances of our system compared to older devices. Interestingly, SPDM/deconvolution imaging showed no to very little axial “smearing” of fluorescence signals in 50 µm deep tissue volumes analyzed (Figure 4B, enlarged insets).

### 3.3. Three-Dimensional Super-Resolution Imaging of Mouse Brain Tissue

The previous CLSM-HyVolution experiments already delivered images with improved spatial resolution over conventional optical microscopes (roughly a factor of 1.5 to 2), and HyVolution may therefore be regarded as a super-resolution technique [43]. In the next steps we aimed at employing other super-resolution microscopy approaches. Using the same tissue specimen as in the previous experiments we next compared the performance of the three super-resolution approaches Airyscan, Lattice-SIM and STED microscopy (see [26,57] for detailed descriptions of the technology of these super-resolution methods) on adjacent regions of the same tissue section. For direct comparison, again identical volumes of the tissue (65 × 65 × 50 µm^3^) were acquired on each microscope platform.

The Airyscan approach (as realized with Zeiss Airyscan) is based on a point-scanning CLSM and through an array or area detector using the principle of structured illumination with dedicated image processing to achieve a flexible improvement in spatial resolution with a factor of 1.5 to 1.8 [58,59]. The Airyscan image of our tissue sample is shown in Figure 5A1. One single optical section after image processing is depicted in Figure 5A2. All fluorescence channels displayed a high SNR without any background blur. The already high image quality was further increased by deconvolution of the image stack (Figure 5A3–A6).

Next, Lattice-SIM microscopy was performed on the brain tissue section (Figure 5B1). Lattice-SIM (as for example realized on the Zeiss Elyra 7) is based on classical SIM to reach an improved image contrast and an up to 2-fold improvement in spatial resolution by large-field illumination of the sample area with a Lattice spot pattern and fluorescence detection on camera [60,61]. Several images (with different Lattice spot pattern positions) have to be recorded and processed to create the final image. Similar to Airyscan, processed Lattice-SIM optical sections displayed all fluorescence signals specifically and with high SNR even without deconvolution (Figure 5B2–B6). A striking advantage however of the camera-based Lattice-SIM was that image stack acquisition and processing was finished in 22 min, whereas the recording of the same sample with Airyscan microscopy took 1.5 h alone (Table 1).

Finally, we applied STED super-resolution microscopy (Figure 5C1). Based on conventional CLSM, STED microscopy achieves an in theory unlimited improvement in spatial resolution (in principle without the need for further image processing) by addition of a red laser with a donut-shaped focal intensity profile to reduce the effective size of the effective fluorescence excitation scanning spot [62]. Yet, final image contrast and resolution may be improved by additional post-processing through deconvolution [63,64]. While the unprocessed STED microscopy images showed some degree of blur (data not shown), deconvolved optical sections revealed protein distribution patterns at nanoscopic resolution (Figure 5C2–C6). This became most obvious for the synapses, which appeared much better resolved in STED compared to Airyscan or Lattice-SIM images.

On the other hand, acquisition time for the STED microscopy image stack was 5.5 h. Additionally, in our case a deconvolution time of more than 3 h was added (Table 1). Thus altogether, STED microscopy-based image production in our case took 23times longer than Lattice-SIM until the final processed image stack was finished (Table 1). Yet, STED, when optimized for single section analysis as compared to our multi-section stacks, offers the chance of deeper insights into sub-synaptic compartment composition and organization [22,65].

To further assess imaging performance, we analyzed single synapses obtained from the different microscopes in orthogonal sections (Figure 6). Consistent with the previous observations (Figure 3 and Figure 5), SIOS performed rather poorly in resolving Homer/Shank2-labeled synapses in z direction (Figure 6, SIOS). Synapses were much better resolved in axial direction using SPDM followed by deconvolution, while best results were obtained by all super-resolution approaches tested here (HyVolution, Airyscan, SIM and STED) (Figure 6).

### 3.4. Approaches to Increase Performance in 3D Tissue Imaging

#### 3.4.1. Leap Mode Virtual Reconstruction in Lattice-SIM Microscopy

We also tested recently introduced commercially available hardware and software improvements in the light microscopy field on our mouse tissue sample. For Elyra 7 (Zeiss) microscopes, a new acquisition and processing mode—called Leap mode–was introduced. The Leap mode allows to reconstruct z planes within the depth of focus around the sampled position in analogy to and advancement of the previously developed ‘*thick Slice* Blind-SIM’ algorithm developed by Jost et al. [66]. This digital sectioning reduces the number of images required to generate a given 3D image stack by a factor of three. Briefly, each camera image is a projection of light along the *z* axis encoded by the point spread function. The illumination pattern along z also modulates the intensity detected in the projection when the sample is shifted laterally. Thus, the SIM phase images from a single position in z also contain information from adjacent planes and can be used for virtual reconstruction of 3D information outside of the optical section. By unmixing the various contributions of adjacent z planes additional z planes can be virtually reconstructed in Leap. This allows to reconstruct 3D data from a series of 2D raw images.

To assess the performance of the Leap mode, every 3rd optical section of a 376 slices-comprising Lattice-SIM z stack (Figure 7A) was extracted to compile a new stack consisting of only 125 optical slices (Figure 7D). Conventional SIM reconstruction of the 376 slices-stack revealed the typical distribution of MAP2, Homer, and Shank2 at super-resolution quality (Figure 7B,C). Strikingly, the Leap reconstruction of the very same optical section (which is not available in the data set) was remarkable similar, if not identical, to the SIM-processed 2D slice (Figure 7, compare C and F). Thus, in Leap mode, only one-third of information was sufficient to produce a complete 3D reconstruction of the 4-color mouse brain tissue section without loss of nanoscopic details. Since computational Lattice-SIM reconstruction is more elaborate than conventional SIM, it took longer to process the image stack, resulting in an overall imaging/processing time of 29 min (Table 1).

#### 3.4.2. Motorized Correction Collar Objectives

Deep tissue imaging can be obstructed by optical aberrations induced by deformations in the wave front of the illumination or fluorescence light due to refractive index (RI) mismatch between the immersion and sample medium, RI changes within the tissue (specimen inhomogeneities), temperature changes, and inappropriate cover glass thickness [67]. Motorized correction collar objectives have the potential to minimize spherical aberrations and thus enable optimized imaging with respect to cover glass variation, resolution, SNR and penetration depth [68]. The correction collar allows adjustment of the central lens group position within the objective mainly to coincide with fluctuations in cover glass thickness. To assess the effect of objective correction in the brain tissue, we performed HyVolution imaging of Homer, Shank2 and DAPI at a penetration depth of ~40 µm (Figure 8A). The same optical section was imaged sequentially with increasing values of the motorized position of a 93× correction collar objective (motCORR, Leica). Figure 8B shows that Homer fluorescence signal intensity within a single synapse varies with the position of the correction collar. Furthermore, at collar positions above 50%, the morphology of the fluorescent signal originating from the synaptic anti-Homer immunostaining became obstructed (Figure 8B).

Signal intensities peaked between positions 15% and 25% of the motorized correction position as judged from the quantitative evaluation. At these collar positions the strongest signal intensities were 2–3 times higher than the lowest (Figure 8C). Thus, motorized correction collar objectives substantially improve intensity and SNR of fluorescence signals in brain tissue.

#### 3.4.3. Adaptive Optics (AO)

Another remedy for optical aberrations is to make use of adaptive optics (AO) [69]. AO elements include deformable mirrors or spatial light modulators. These elements have the potential to counterbalance optical aberrations by a compensating wave front deformation outside the sample within the microscope [70]. For example, previous use of AO has shown significant improvements in contrast and resolution on 3D STED microscopy images of tissue samples [39]. Following our previous AO STED microscopy approach using a spatial light modulators [37], we recorded STED images of Homer improved with AO in the same immunolabeled brain tissue samples as before (Figure 8D). We chose Homer since a red-emitting label (gp-anti-Homer-STAR 635P) fitted the excitation and detection lines of our microscope. Clearly, changing the wave front of the bottle-shaped laser beam improved the contrast, reduced the background haze, and improved the spatial resolution, as highlighted by the profiles along the lines in-between the arrows in Figure 8D. Most importantly, the improvement was more dominant for deeper sections, highlighting the expected increase of optical aberrations and impact of AO with sample depth.

#### 3.4.4. CUDA-GPU Accelerated Image Processing

Finally, we approached the issue of accelerating 3D image processing time. As highlighted in Table 1, processing time is reaching minutes to hours, depending on the amount of algorithms implemented such as deconvolution. One way to acceleration is to parallelize processing. CUDA is a programming technique, with which highly parallelized computational program streams can be processed on a graphics card (GPU). Commercially available CUDA-GPUs therefore are versatile tools to accelerate processing of imaging data, in particular deconvolution [71]. To assess CUDA performance, we applied deconvolution of 4-color confocal image stacks derived from the brain tissue section. The confocal data sets contained increasing numbers of z-sections, mounting to data files ranging from 48 MB to 1536 MB. Deconvolution of these files was done with and without CUDA support. This analysis showed that CUDA-GPU acceleration is moderate for small data files but becomes progressively more accelerated with larger file size (Figure 8E).

## 4. Discussion and Conclusions

By imaging the very same multi-fluorescently stained brain tissue slice we have compared several commercially available microscopy platforms at various size scales. We hope that the data collected will help both researchers and microscopy facility managers to select the right tool to address specific biological questions in (brain) tissue imaging.

We would like to point out that most of our observations are qualitative. In addition, many additional aspects must be considered when selecting the right imaging tool(s) to address specific biological questions. These include (i) the choice of tissue/cell embedding medium and objectives (water, oil, or glycerin) to obtain refractive index match, (ii) the choice of detector (photomultiplier, gallium-arsenite, hybrid) to obtain the best possible SNR, (iii) fine tuning of the confocal pinhole to optimize photon yield vs. resolution, (iv) application of various deconvolution algorithms along with application of measured instead of theoretical point spread functions, (v) the pixel dwell time of the scanning laser (e.g., employing novel resonant beam-scanning systems), and (vi) PC power for accelerated image acquisition, processing and post-processing. Finetuning of imaging conditions according to these parameters would certainly have improved the image quality obtained in our study. With respect to imaging speed it will also be interesting to test fast confocal scanners, such as resonant galvanometer scanners [72].

While some of our observations may have been predictable, others were not anticipated and interesting to learn about. For example, the spinning disc confocal microscope delivered impressively high-quality images after deconvolution. Well-known for its high acquisition speed in live cell imaging, the SPDM in combination with image deconvolution is obviously also very well suited for rapid 3D tissue imaging (Figure 3B and Table 1). The detection of Homer and Shank2 in individual synapses (Figure 3B5) would allow for quantification of synapse markers in larger parts of the brain at reasonable time scales. More sensitive cameras (than the one we used) and online deconvolution (see below) may accelerate imaging speed even further. Altogether, based on our observations, the SPDM in combination with deconvolution is a highly competitive alternative to point scanning confocal systems with respect to image resolution and certainly superior with respect to imaging speed (Table 1).

In STED microscopy imaging bleaching of fluorescence during z-stacks (data not shown) forced us to reduce the additional depletion laser power to levels that allowed lateral resolution in the 70 nm to 90 nm range only. Although the lateral resolution of Lattice-SIM is less (d = 120 nm) and although 13 grid images are recorded for each optical section, this technique is much better suited to image the 65 × 65 × 50 µm^3^ tissue cubes we used for comparing the different techniques at super-resolution with only little bleaching in our mouse tissue sample. Moreover, the acquisition/processing time in Lattice-SIM was in our case at least one order of magnitude faster than in STED (Table 1). To improve the STED image acquisition in 3D, we will in the future test novel low-illumination algorithms such as the DyMIN, RESCue and MINFIELD STED imaging modes [73]. Additionally, STED is straightforwardly further improvable using novel technology such as AO (Figure 8D) or improved objectives with different immersion media than oil [38].

Our study clearly highlighted again that deconvolution is a powerful tool to improve image quality and resolution in widefield (Figure 2C2d,3A5), spinning disc confocal (Figure 3B5), point scanning confocal (Figure 3C5), and STED (Figure 5C) microscopy. We therefore highly recommend to apply deconvolution of microscope images whenever possible. Confocal images increase in quality when averaging is activated during scanning. However, line averaging can be accompanied by a loss of information (smoothing-out of structures). We noted that averaging was not necessary in STED when deconvolution was performed on non-averaged optical sections (data not shown). Therefore, deconvolution is also a tool to avoid averaging-induced photobleaching of the sample. Deconvolution as well as SIM processing, and in particular the time-consuming Lattice-SIM Leap processing, markedly contributed to the overall time until the final image stack is finished (Table 1). This issue has been addressed already in HyVolution confocal super-resolution microscopy (Figure 3C and Table 1) by parallelizing image acquisition (CPU) and deconvolution (CUDA-GPU) [41]. Hence, in the future automatic online processing should be implemented in commercial microscope software, as for example in LIGHTNING, which recently replaced the HyVolution 2 module on Leica platforms.

The tested techniques represent state-of-the-art methods in microscopy that are commercially available and therefore directly accessible for the use in biological research. Recently, several potentially highly attractive approaches have been described that once turned into commercial devices also hold great potential to be used in brain research. With respect to imaging speed in 3D, certainly light sheet fluorescence microscopy (LSFM) would have outperformed all other platforms used here [74]. LSFM offers optical sectioning capability and allows for imaging of biological samples with reduced background and photobleaching [75]. An improvement in image contrast and resolution was realized with Lattice light sheet microscopy (LLSM), which appears to be an excellent tool in high resolution 3D imaging of fixed tissue up to 20 µm thick [76] or in expanded biological samples [77]. A combination of single-molecule localization microscopy with light sheet illumination may allow extremely rapid 3D super-resolution microscopy [78]. Cleared tissue axially swept light-sheet microscopy (ctASLM) has recently been developed for super-resolution imaging of spines in tissues [79]. However all these highly sophisticated approaches are not yet available on commercial microscopes. The recently introduced Thunder Imagers (Leica) are camera-based fluorescence microscopes tailored to eliminate the out-of-focus blur in thick specimen by ‘computational clearing’, a new opto-digital method. Based on the approach reported here, it will be exciting to conduct a side-by-side comparison of the new techniques for their practical usability in brain research.

In general, the demands of imaging of brain tissues as in biomedical research are large, with applications requiring different aspects of spatial and/or temporal resolution, 3D imaging deep inside the sample, long acquisition times with low photo-toxicity, and/or extensive data processing. The fact that all microscopy approaches are complementary, whether they are diffraction-limited or with nanoscale resolution, promotes research environments with access to various kinds of microscopes and nanoscopes, depending on their suitability for the case in hand. On the other hand, research environments should be strongly interdisciplinary, allowing (bio-)chemists, physicists, engineers and biomedical researchers to tightly work together to optimize technology for its proper use in aging-related and biomedical research. An ultimate goal would be a microscope that can combine all of the demands set by the users. Unfortunately, much advanced technology developed by engineers, physicists, or chemists has not made its way into applications, simply because of missing tight links with biomedical users. Therefore, possibilities have to be created, promoting interdisciplinary research environments with access to a broad range of complementary state-of-the-art technology as well as with the chance to test out new approaches. Many newly introduced commercial microscope platforms testify to this development, including Minflux [80], Super-Resolution Spinning Disc (SoRa) microscopy [81], single-molecule based Nanoimager ONI [82], Abberior STED facility line [83], Abbelight SAFe 180/360 [84], nanoFLeye [85], CODIM imager [86] or Re-scan microscopy platforms [87]. It will be interesting to see how these new machines will perform in tissue imaging as employed here.

## Figures and Tables

**Figure 1 cells-09-01377-f001:**
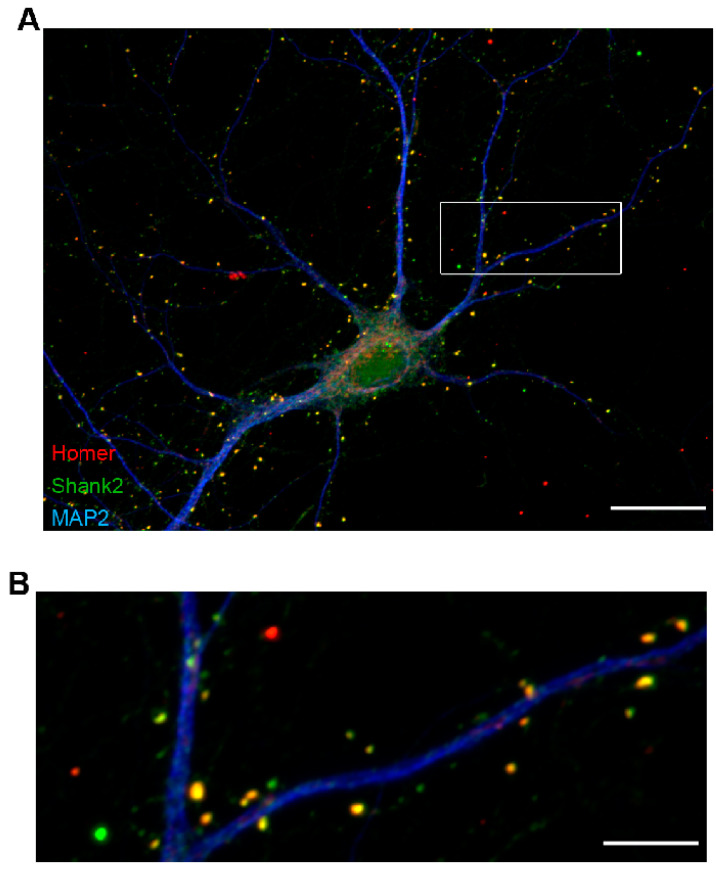
Single cell layer representation of postsynaptic proteins Homer and Shank2/ProSAP1. (**A**) Immunostaining of cultured rat hippocampal neurons (DIV16) recorded with a 63×/1.4 objective on a Zeiss AxioObserver.Z1 equipped with an ApoTome.2 confirms colocalization of the postsynaptic density proteins Homer 1 (Homer; red) and postsynaptic density protein Shank2/ProSAP1 (Shank2; green) at structures protruding from dendrites marked by microtubule-associated protein 2 (MAP2; blue). (**B**) Enlarged view of the boxed region depicted in A. Scale bar, 20 µm in (**A**), 5 µm in (**B**).

**Figure 2 cells-09-01377-f002:**
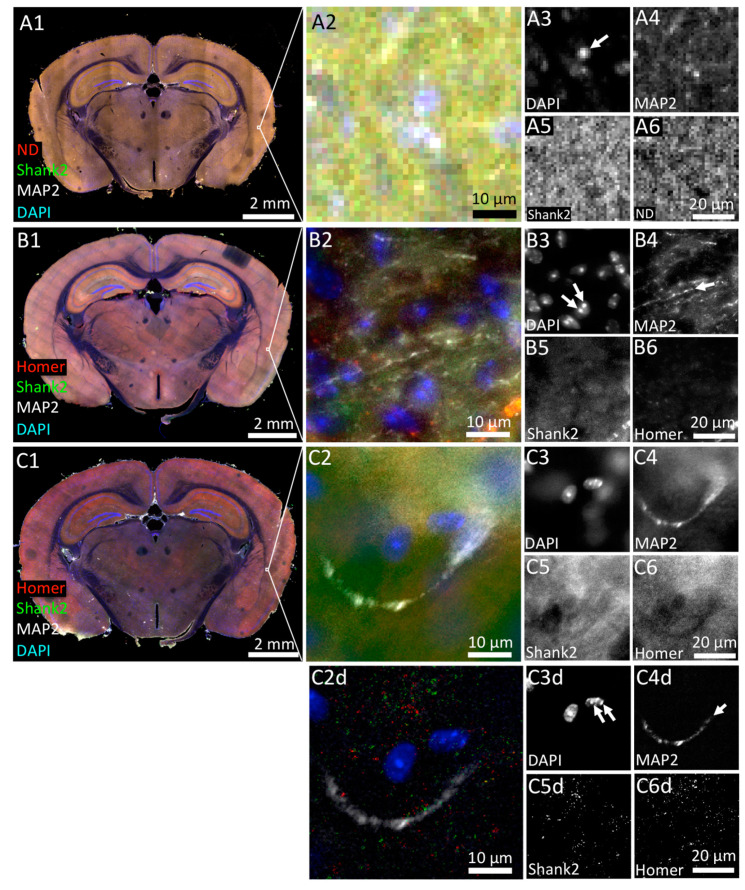
Comparison of widefield microscopy methods in the imaging of tissue sections. Mouse brain sections (60 µm thick) immunofluorescently labeled with antibodies against Homer1 (Homer; red in **B** and **C**), Shank2/ProSAP1 (Shank2; green) and MAP2 (white) imaged by 3D-Zoom microscopy (**A**), a slide scanning microscope (**B**) and a conventional widefield microscope (**C**), respectively. DNA was counter-stained with DAPI (blue). Note that fluorescence signals of Homer staining in (**A**) were not captured due to lack of an appropriate fluorescence excitation. Instead, an additional place holder fluorescence channel was captured (ND, red) to allow later comparison of acquisition speed with the other widefield methods (see Table 1). (**A2**) Magnified view of a cortical area boxed in (**A1**). Individual immunofluorescence signals of this subregion are shown in monochrome in **A3** to **A6**. An arrow in A3 depicts an example of a cell nucleus. (**B1**) The same sample as in A imaged in 3D using a multi-slide scanning microscope. (**B2**) Single optical section shown as a magnified view of the subregion boxed in **B1**. (**B3** to **B6**) Individual immunofluorescence channels of the image in **B2** in monochrome. Arrows in B3 and B4 mark individual chromo-centers within single nuclei and MAP2-positive dendrites, respectively, and thereby highlight an improved resolution provided by the multi-slide scanning microscope. (**C1**) The same sample imaged in 3D using a conventional widefield microscope. (**C2**) A single optical section was selected from the dataset shown in **C1** (marked with a white box) and split into individual fluorescence channels (**C3** to **C6**). Images **C2d** to **C6d** show the same subregion after deconvolution. Individual chromo-centers within single nuclei as well as MAP2-positive dendrites become clearly resolvable after deconvolution (Arrows in C3d and C4d, respectively).

**Figure 3 cells-09-01377-f003:**
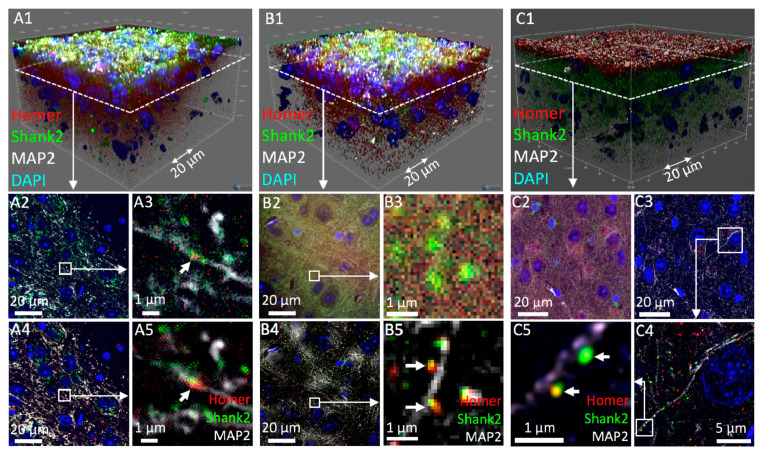
Comparison of various confocal microscopy methods in tissue imaging. Comparison of 3D imaging of a 90 × 90 × 50 (x/y/z) µm^3^ region of a mouse brain section immunofluorescently labeled with antibodies against Homer (red), Shank2/ProSAP1 (green) and MAP2 (white) using SIOS (ApoTome) microscopy (**A**), SPDM (**B**) and conventional point-scanning confocal microscopy with deconvolution (HyVolution) (**C**). **A1**, **B1**, and **C1** show 3D reconstructions of deconvolved image stacks. DNA was counter-stained with DAPI (blue). (**A2**) shows a SIOS-processed single optical section of the image stack recorded in **A1**. A subregion of this optical section (white box) is shown in **A3**. (**A4**) and (**A5**) show the same optical section and subregion, respectively, after deconvolution in ZEN software. Arrows in (**A3**) and (**A5**) indicate potential identification of a single synapse at a MAP2-positive dendrite. (**B1**) The same tissue sample as in (**A1**) imaged at an adjacent position using a spinning disc confocal microscope. (**B2**) shows a single confocal section of this image stack. (**B3**) shows a subregion of the same section. (**B4**) and (**B5**) show the same optical section after deconvolution in ZEN software. Arrows indicate identification of Homer/Shank2-positive synapses at dendrites. (**C1**) The same specimen used in **A1** and **B1** was subjected to conventional point-scanning confocal microscopy but with the pinhole set to 0.5 Airy units. Furthermore the entire image stack was subjected to deconvolution by Huygens software within Leica’s HyVolution module. (**C2**) shows a single optical section of the original image stack. (**C3**) represents the same optical section shown in **C2** after deconvolution. A subregion of the deconvolved section was selected for enlarged views (**C4** and **C5**). Arrows in **C5** indicate identification of Homer/Shank2-positive synapses at dendrites.

**Figure 4 cells-09-01377-f004:**
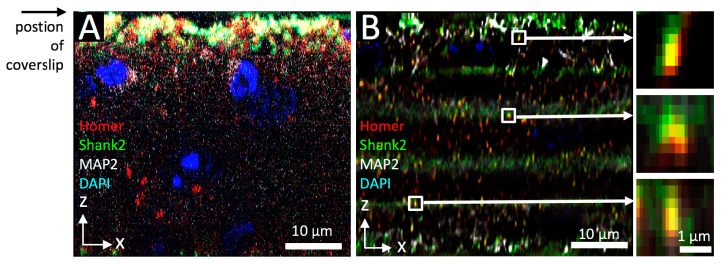
Comparison of optical sectioning and confocal/deconvolution microscopy in synapse imaging of mouse brain tissue. Image stacks of mouse brain labeled to detect Homer (red), Shank2/ProSAP1 (green) and MAP2 (white) were acquired with SIOS (ApoTome) microscopy (**A**) or SPDM/deconvolution (**B**). DNA was counterstained with DAPI (blue). Images show orthogonal views of each image stack. Single accumulations of colocalized Homer and Shank2 fluorescence, likely representing PSDs, were selected (white box in **B**) and shown as enlarged x/z views on the left side of **B**.

**Figure 5 cells-09-01377-f005:**
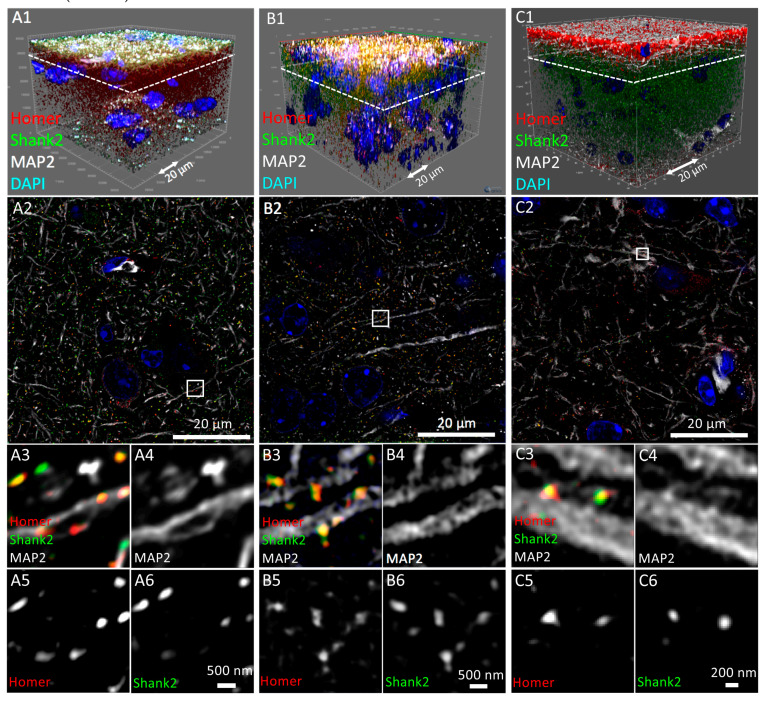
Comparison of various super-resolution microscopy methods in tissue imaging. A 90 × 90 × 50 (*x*/*y*/*z*) µm^3^ 3D region of mouse brain immunofluorescently labeled with antibodies against Homer1 (Homer; red), Shank2/ProSAP1 (Shank2; green) and MAP2 (white) was imaged using an array detector (Airyscan) microscope (**A1**), a Lattice-SIM super-resolution microscope (**B1**) and a STED super-resolution microscope (**C1**), respectively. DNA was counter-stained with DAPI (blue). Please note that **A1**, **B1**, and **C1** show 3D reconstructions of deconvolved image stacks. (**A2**) shows a single optical section of the **A1** image stack after processing and deconvolution. A subregion of this image slice (white box in **A2**) is shown enlarged as a merged view as well as in monochrome individual channels (**A3**–**A6**). (**B1**) The same tissue sample used in **A1** was imaged at an adjacent position employing Lattice-SIM super-resolution microscopy. (**B2**) shows a single optical section of the image stack shown in B1. A subregion of this section was selected (white box) for display as enlarged views of the merged as well as individual channels (**B3**–**B6**). (**C1**) The same tissue sample used in **A1** and **B1** was imaged at an adjacent position employing STED super-resolution microscopy. (**C2**) shows a single optical section of the image stack shown in **C1** after deconvolution. A subregion of this section was selected (white box) for display as enlarged views of the merged as well as individual channels (**C3**–**C6**). Note the higher resolution of the STED microscopy (bar in **C6**, 200 nm instead of 500 nm, as in **A6** and **B6**).

**Figure 6 cells-09-01377-f006:**
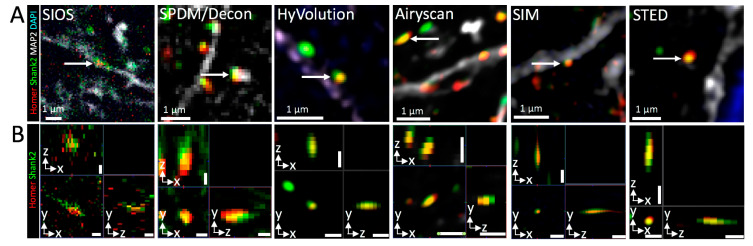
Comparison of various microscopy methods in 3D synapse imaging of mouse brain tissue. (**A**) shows single optical sections of the image stacks described in Figure 3 and Figure 5 ((Mouse brain immunofluorescence labeling of Homer1 (Homer; red), Shank2/ProSAP1 (Shank2; green) and MAP2 (white); DNA counterstained with DAPI (blue)). Single focal accumulations of Homer and Shank2/ProSAP1, likely representing PSDs were selected from each section (white arrows) and displayed as orthogonal views in **B**. SIOS, structured illumination optical sectioning (ApoTome); SPDM, spinning disc confocal microcopy; SIM, structured illumination microscopy; STED, stimulated emission depletion. Please note, that 2D-SIM and 2D-STED was employed here, while the HyVolution and Airyscan stacks were deconvolved resulting in improved z-resolution. Bars, 1 µm (**A**) and 500 nm (**B**).

**Figure 7 cells-09-01377-f007:**
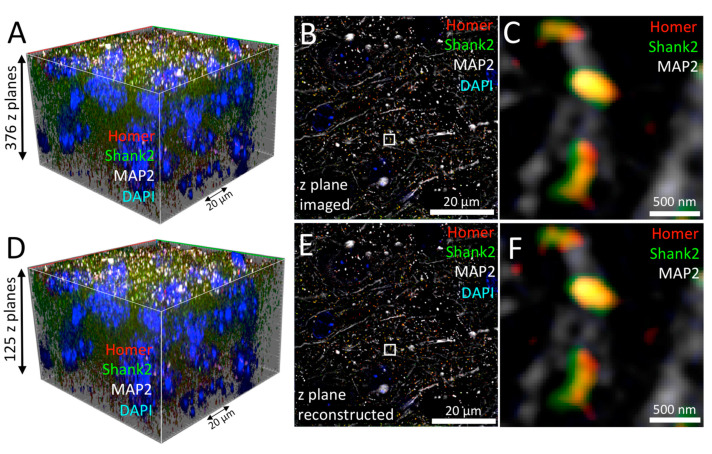
Lattice-SIM Leap reconstruction in tissue imaging. (**A**) A 90×90 × 50 (*x*/*y*/*z*) µm^3^ 3D region of mouse brain immunofluorescently labeled with antibodies against Homer (red), Shank2/ProSAP1 (green) and MAP2 (white) was imaged using Lattice-SIM microscopy. DNA was counter-stained with DAPI (blue). A single optical section of the image stack is shown in **B**. (**C**) Enlarged view of a selected region (white box) of the image in **B**. (**D**) Leap reconstruction of the dataset shown in (**A**) after extraction of only every 3rd optical section. (**E**) Single optical section after reconstruction applying the Leap algorithm. (**F**) Enlarged view of a selected region (white box) of the image in **E**.

**Figure 8 cells-09-01377-f008:**
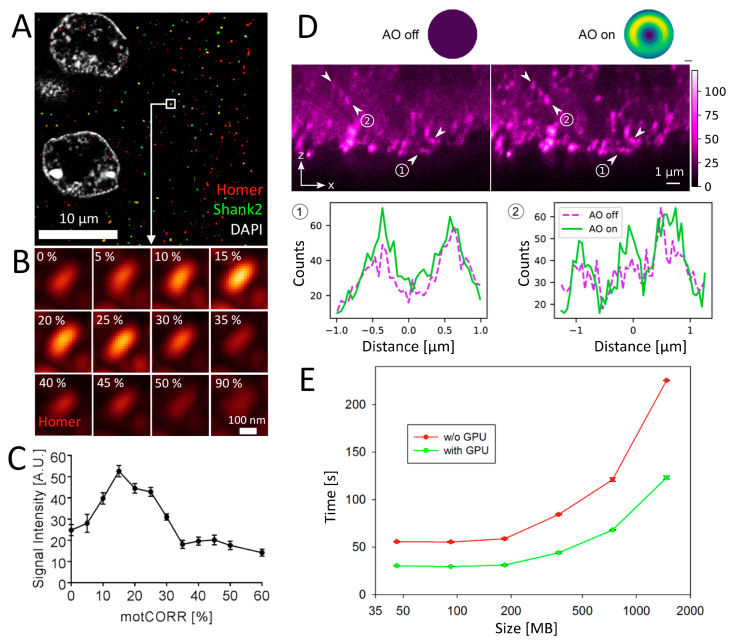
Methods to improve deep tissue imaging and image processing. (**A**–**C**) Influence of an objective correction ring on image quality. Mouse brain tissue immunofluorescently labeled for Homer (red) and Shank2 (green). DNA was counterstained with DAPI (white). Imaging was performed in an area 40 µm deep within this tissue section employing point-scanning confocal microscopy (**A**). The same optical section was imaged sequentially with increased settings of a motorized correction ring of a 93× glycerol objective. The signal intensity at a selected synapse of Homer fluorescence (boxed in **A**) dependent on correction ring setting is shown in **B**. (**C**) Quantitation of signal intensity vs. motorized correction ring setting (motCORR) as shown in **B** (*n* = 5 per data point). (**D**) Effect of AO: Representative z-STED microscopy images of Homer (magenta, gp-anti-Homer-STAR 635P immunolabeled) in the mouse brain tissue without (left, AO off) and with (right, AO on) AO correction on the bottle-shaped STED laser (added wavefront distortion in the upper right insets, arbitrary color scale from 0 (blue) to maximum (yellow)) with intensity profiles along the lines in-between the arrows. Scale bar, 1 µm. (**E**) Acceleration of deconvolution by CUDA graphics card. Image stacks of various thickness of the mouse brain tissue as shown in **A** were acquired on a confocal microscope. Deconvolution of the stacks was performed with or without employment of a CUDA graphics card. The time required for deconvolution was measured (*n* = 3) and plotted versus the file size of the image stack.

**Table 1 cells-09-01377-t001:** Comparison of different microscopy platforms used to image the same specimen.

Microscope		Stereo	Slide Scanner	Convent. Widefield	SIOS (Apotome)	SPDM	Confocal (Hyvolution)	Airyscan	Lattice-SIM	Lattice-SIM Leap	2D-STED
Objective		2.3×	20×	20x	63× Oil	63× Oil	63× Oil	63× Oil	63× Oil	63× Oil	100× Oil
Sample Size [µm]	x	1 × 10^4^	1 × 10^4^	1 × 10^4^	90	90	90	65	65	65	65
	y	0.8 × 10^4^	0.8 × 10^4^	0.8 × 10^4^	90	90	90	65	65	65	65
	z	50	50	50	50	50	50	50	50	50	50
Optical Sections		8	16	80	210	197	448	273	376	125	276
Voxel Size [nm]	x	n.a.	n.a.	n.a.	93	102	35	40	31	31	31
	y	n.a.	n.a.	n.a.	93	102	35	40	31	31	31
	z	n.a.	n.a.	n.a.	240	240	123	110	110	110	183
Theor. Resolution [nm]	x	n.a.	n.a.	n.a.	220	220	140	120	100	100	40
	y	n.a.	n.a.	n.a.	220	220	140	120	100	100	40
	z	n.a.	n.a.	n.a.	600	600	380	350	300	300	600
File Size		2.7 GB	16.6 GB	840 GB *^3^	11 GB	1.8 GB	12 GB	69 GB	42 GB	14 GB	21 GB
Acquisition Time		40 min	1 h 15 min	10 h 52 min *^3^	18 min 25 s	5 min 38 s	3 h 42 min	1 h 38 min	12 min 44 s	4 min 17 s	5 h 26 min
Processing Time *^1^		2 min 9 s	0 min *^2^	51 h 12 min *^3^	2 min 30 s	n.a.	n.a.	54 min	9 min 13 s	24 min 40 s	n.a.
Deconvol-ution Time		n.a.	n.a.	n.a.	8 min 16 s	5 min 24 s	0 min *^2^	16 min 3 s	n.a.	n.a.	3 h 20 min
Total Time		42 min 9 s	1 h 15 min	62 h 2 min *^3^	20 min 55 s *^4^ 26 min 41 s *^5^	11 min 2 s	3 h 42 min	2 h 48 min	21 min 57 s	28 min 57 s	8 h 46 min

*^1^ Processing time to calculate extended depth of focus (Zoom), stitching of tiles (Zoom, Slide Scanner, Widefield) or SIM reconstruction (Structured Light, Lattice-SIM, Leap). *^2^ Processing is required but occurs during acquisition. *^3^ Numbers were extrapolated from imaging, processing and deconvolution of a subregion. *^4^ time required if SIM reconstruction is applied without deconvolution. *^5^ time required if deconvolution is applied without SIM reconstruction. SIOS, Structured Illumination Optical Sectioning, SPDM, Spinning Disc Confocal Microscopy; SIM, Structured Illumination Microscopy; STED, Stimulated Emission Depletion; GB, Gigabyte.

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
