# Peer review of "Comparison of Multiscale Imaging Methods for Brain Research"

_cells, 2020, doi:10.3390/cells9061377_

Round 1

Reviewer 1 Report

Light microscopy is an important tool in the study of brain organization and how this changes with aging or due to disease. The challenge for researchers studying these processes lies in using microscopy to quantify these – often subtle – changes. With the very fast-paced development of novel light microscopy techniques, this study by Tröger et al aims to provide a comparison of different light microscopy techniques to study mouse brain slices across different scales, by focusing on widely available (commercialized) methods.

This study provides a nice overview that will be helpful for a brain researcher to decide which microscope to use for their research, including very useful comparisons on the requirements for image acquisition and the necessary processing (time and computation) for each different method. It also highlights very well how useful deconvolution can be, a method that is probably underused by many researchers to get the most out of their images, even when confocal microscopy is used.

Drawbacks to this study in my opinion are the fact that the results presented here are relatively unsurprising, and secondly, that it risks being outdated very quickly. While commercialization takes a lot more time than the quick pace of technical advancements and the emerging of new methods, it does also move quite fast these days. For example, the authors include the Zeiss Elyra 7, which (to my knowledge) has not been available for very long and is probably not up-and-running in that many places at the time of writing. At the same time the authors mention in their discussion the even newer Leica Thunder system, which indeed would be very useful to include in a study such as this. Similarly, improved methods for lightsheet imaging are become available incredibly fast and would also deserve a place in a comparative study such as this one. I’m aware, especially given the current circumstances, adding new data is not feasible. Hence, I’ve attempted to look at the data that’s available and find ways in which this study would be of higher interest and remain relevant even when newer techniques (such as the Thunder system) become more widely available. In this regard, the main aspect that is missing from this manuscript in my opinion is a quantitative comparison of the resulting images achieved using the different methods. A more thorough, quantitative description of the final images would allow the reader to get a better idea of where the limits of each method lies. This would be especially useful to help a researcher decide at which point super-resolution imaging becomes a necessity. In addition, it could theoretically allow researchers to compare their own images, obtained on other/newer systems, with the results obtained in this study. Below I have some suggestions which I think will improve the paper.

General comments:

A big part of the introduction mentions the effects of aging on the brain, particularly changes in (post-)synaptic structure, and the need for imaging methods to quantify these changes. I do really appreciate the effort made here to showcase the different imaging techniques at their best, but I do miss how this links back to performing quantitative studies. In an ideal world, this paper would include a comparison of a “young” brain vs an “aged” brain, to clearly show how the different methods perform in such a study, but I’m aware this is beyond the scope of this paper. Still, what I’m mostly missing from looking at the data as it’s presented here, is the added benefit of the super-resolution methods, when the confocal methods are clearly sufficient for the identification of these PSDs with high confidence. Some suggestions:

  • It would be great to get some quantitative data out of each imaging modality and compare the different confocal and super-resolution methods. I’m not a brain researcher so I’m not sure what the major changes in these structures are due to aging, but any quantifications on number or morphology of PSDs or the MAP2-positive dendrite structure could be of interest to compare the methods and/or showcase their limitations.
  • Table 1 is a great resource, clearly showing the trade-offs in terms of imaging time and computational power needed to acquire a certain volume for each technique. It would be very useful to also supplement this with some “quality” measures (achieved lateral and axial resolution, contrast, …), to give the reader an idea of what the added cost of confocal or super-resolution imaging is buying them at the end of the day (rather than just a “prettier picture”).

Figure 3:

  • the way shown in A is more informative: with and without deconvolution for both a large and a zoom area. In B and C, we can’t really judge the deconvolution effect very well on the final fine-detail. It would be better to show it the same way as in A.
  • The same brain region is being imaged and while the 2D slices shown look pretty similar, there seems to be a significant difference when looking at the volume data in terms of distribution of the different markers of A1 and B1 vs C1. Is this a difference of processing and display, or could that be caused by artefacts such as photobleaching in the LSCM?
    • In case it is photobleaching, it could be interesting to compare the top and bottom slices in SDCM vs LSCM.

Figure 4:

  • As for Fig. 3, in Fig. 4 A1 vs B1 vs C1 look surprisingly different in terms of distribution of the different markers throughout the volume. What’s the cause here?

Minor comments:

  • Check the affiliations, 5 and 6 are not linked to any of the authors
  • Since the introduction mentions that novel methods have outpaced commercialization, a reference to such studies would be relevant for the discussion of where commercialization is versus where the forefront of tech development is. Gao et al, Science 2019 (PMID: 30655415) comes to mind as a very relevant reference here.
  • In the methods section, include the reference to the original ImageJ or FIJI (whichever is appropriate) papers (line 173).
  • Line 229: there’s an f is missing in “fluorophores”
  • In the section on grid confocal microscopy, it would be good to include some of the limitations of this technique in the text, as it’s quite relevant for the reader. An issue that comes to mind immediately is slice thickness, which for grid confocal is ideally around 20um (https://www.ncbi.nlm.nih.gov/pmc/articles/PMC4365987/). Have you noticed any of these issues with your slices?
  • I couldn’t find reference 73 and 74 back in the text. Please check all references again.

Reviewer 2 Report

Troeger et al present a comparison of commercially available microscopy methods to image fixed brain sections in 3D, using widefield, structured illumination, confocal and STED microscopy, and the application of deconvolution microscopy. This includes structured illumination for optical sectioning (Apotome) and for super-resolution (Lattice-SIM), different variants of confocal microscopy (spinning disk, sub-airy pinhole and Airy scan) and two variants of STED microscopy (systems from Leica and Aberrior, the latter is also used to demonstrate adaptive optics). The same brain slice was imaged with each method, and the resulting image data is compared. Among other things, spines are resolved by some of the techniques and pre and post-synaptic markers are observed. The different imaging techniques are also compared in their acquisition and post processing time, and data volume.

The authors find that a spinning disk microscope, combined with deconvolution allows relatively rapid volumetric image acquisition and good optical sectioning capability. The STED microscope is noted for its highest resolving power.

I think this is a timely and helpful review for neuroscientists that have to navigate the ever increasing number of available microscope techniques. The study is pretty comprehensive, and the one-to-one comparisons could prove valuable to end users.

Some concerns arose about resolving power and 3D imaging performance, which was only addressed qualitatively. Also, the prospect of tissue clearing was not mentioned in this article, although this has become an important method to allow high resolution imaging in brain tissues, enabling among other things 3D imaging of spines (Chakraborty, Tonmoy, et al. "Light-sheet microscopy of cleared tissues with isotropic, subcellular resolution." Nature methods 16.11 (2019): 1109-1113.). I was surprised in this regard that the authors used PFA fixed samples, which may be even more scattering than native brain tissue.

As for the resolution, some more quantification would be appreciated. Fourier ring correlation could be a possible way to evaluate the resolution in the tissue samples. However, FRC requires two consecutive images of the same area, which would be unreasonable to request now for a revision. But maybe adjacent z-slices could be used if sufficiently fine z-sampling was used. Alternatively, there is a method to estimate the resolution based on a single image, which showed promising results in our hands (Descloux, Adrien Charles, Kristin Stefanie Grussmayer, and Aleksandra Radenovic. "Parameter-free image resolution estimation based on decorrelation analysis." Nature methods 16.ARTICLE (2019): 918-924.).

For the 3D imaging performance, it appears that mostly lateral views are shown for comparison (only exception for the AO STED Bottle beam imaging), which is insufficient to assess 3D imaging performance. 3D renderings are shown, but they can be deceiving. I would strongly encourage to show some comparisons of axial cross-sections (or local maximum intensity projections, i.e. X-Z or Y-Z views), which would allow the reader to compare the axial resolution the different methods achieve. The pre and post-synaptic markers, as well as spines, should provide small enough structures to show the axial smearing.

Smaller points

Details on Leica STED were missing:  Was the 775nm laser pulsed? If so, what was the repetition rate and pulse length? What was the depletion pattern, Vortex or bottle beam?

On the Aberrior STED system: the power indication of 82mW ambiguous, it also matters what the pulse length and repetition rate is, please indicate those too. Was only the Bottle beam used on this microscope, or a combination of Vortex and bottle beam?

For the imaging with the Apotome system, the authors use the abbreviation SLM. This is an unfortunate acronym, as it reappears for spatial light modulator in the same manuscript, and it is also not consistently used throughout the manuscript. E.g. In figure caption 3, the authors call it “grid-confocal” and later “SIM”.

I personally would recommend to refer to is as structured illumination for optical sectioning and provide the citation by Tony Wilson et al. then it would be clear what it is.

Further questions for Figure 3: Are A1-A3 also deconvolved? Please check if the Zeiss software automatically deconvolves the reconstructed Apotome data.

Figure 3, A4 shows some stripe artifacts, please verify if this is true. Maybe the phase steps were not exactly 2pi/3. This has in principle been solved (Wicker, Kai, et al. "Phase optimisation for structured illumination microscopy." Optics express 21.2 (2013): 2032-2049.), but it might be that Zeiss does not use phase step estimation and assumes that the phase steps are hard-coded. Another possibility is bleach artifacts of the strip pattern that get “imprinted” on the sample, albeit this is more rare.

In the section titled “Spinning disk confocal microscopy”: “luorophores” instead of “fluorophores”.

In the section titled: “Lattice-SIM microscopy”: Please confirm that it is indeed 13 images per focal plane? This seems an odd number. To better understand it, it would be of interest which lattice pattern (5 beam interference?) was used.

Section “3 Results”:  Please provide the voxel sizes used for Nyquist sampling in Table 1. The authors make it sound like they used “critical Nyquist sampling”, i.e. just a factor two smaller than the axial resolution.

I found it interesting, that the Hyvolution technique was left out of section 3.3 “super-resolution”.

Do the authors not consider Hyvolution as a super resolution technique? Is the cross-over point to super-resolution the factor 1.8 provided by Airy scan, and higher (SIM, STED)? I know this is a controversial issue (what is super-resolution), but in an abstraction, both Hyvolution and Airy scan are confocals with a sub-Airy unit pinhole.

Figure 4 mentions “Arrows indicate a single synapse at a MAP-2-positive dendrite. “, however, I could not find any arrows.

Further small question about the data acquisition: Did the order of imaging matter (which technique was used first), or did the authors just image different regions?

LSFM discussion: It should be mentioned that LSFM operates at much lower NA, even the lattice light-sheet system has only an NA of 1.1, which is a rather medium NA. Also sample mounting on a 5mm coverslip causes more challenges for brain slice imaging. The authors may include in their discussion that recent advances in single objective light-sheet microscopy may address these issues. (e.g. https://andrewgyork.github.io/high_na_single_objective_lightsheet/ , effective NA 1.35 and compatible with conventional sample mounting).

Author affiliations: no author seems to be affiliated with institutions 4 and 5? Please revise.

Reviewer 3 Report

In this manuscript Tröger et al. compared different commercially available optical imaging technologies as well as advanced digital image processing methods mainly for aging studies in the (mouse) brain. The authors consider this work as a useful set of protocols and as a guide for scientists in this field to select the appropriate microscopic technique for related applications.

However, I think this manuscript neither includes a protocol section nor can it be used as a reliable guide. In particular, the choice of the microscope setup for each system is highly subjective. Furthermore, critical aspects of applied post-image processing methods were not addressed, obtained findings were not sufficiently discussed and/or related works published elsewhere were not cited. Moreover, some affiliations indicated in the title page of the manuscript were not assigned to co-authors.

Major concerns,

  1. Title page: Affiliation #5 (company) and #6 are not assigned to a co-author(s).
  2. Line 24: Optical imaging of the brain as a complex structure is challenging for several reasons in different model systems (autofluorescence, light scattering in thicker samples …). The rationale why the authors pointed out the aging aspect, which covers a large part of the introduction section, is not clear to me.
    Please comment on this.
  1. Line 27: Quantification of acquired image objects from brain samples is neither performed nor evaluated in this manuscript.
    Please clarify this in the text.
  1. Line 30, 656, 606: The authors did not use large (sample) volumes but rather thin tissue slices of 50-60 µm thickness. They acquired a larger field of views resulting in an increased amount of image data. This is misleading. A large volume would be e.g. an entire mouse brain. Useful technologies for the required deep tissue imaging would be light-sheet microscopy of optically cleared specimens or two-photon microscopy, which were not used in this study.
    Please clarify this in the text.
  1. Line 31: In the microscopy community STORM/PALM or STED are considered as true superresolution techniques; e.g. SIM, Airy scanning, confocal and mathematical solutions to compensate the PSF of the optical system are considered as extended-resolution techniques.
    Please clarify this in the text.
  1. Line 35: There are no protocols shown in this manuscript. A typical protocol is a step-for-step guide through the individual procedure and particularly includes the important note sections where e.g. pitfalls and critical aspects are discussed.
    Please rearrange the corresponding parts of the text.
  1. Line 111: ‘… in recent years …’. E.g. STORM/PALM has a resolving power of ~ lateral 20 nm and axial 50 nm and is used for more than a decade e.g. for studying pre- and postsynaptic scaffolding proteins (e.g. Dani et al. 2010. Neuron 68(5): 843–856).
    Please omit ‘recent years’.
  1. Line 113: What means "extremely" in this context?
  1. Line 118: What is the difference between a point scanner and (sub-Airy) a confocal microscope?
  1. Line 118: Confocal microscopy is the most important technology in life sciences for decades. Why did the author not use a confocal with a pinhole size set to Airy 1? The use of sub-Airy pinhole sizes can be very limited (decrease of SNR). Demonstrating the performance of a confocal under Airy1 conditions would be beneficial for many scientists.
  1. Line 123: CUDA does not ‘improve’ light microscopy; it can accelerate data processing.
  1. Line 153, line 166: I do not think that the used 63x NA 1.4 immersion objective is suitable to image these specimens. Both Fluoromount-G and Mowiol has a RI of ~1.4. The immersion oil required for the high numerical aperture lens is RI:~1.52. The RI index mismatch leads to strong aberrations, which increase with the depth of the sample (starting from ca. 10 µm thickness). That is the reason why typically water immersion lenses or in this case, glycerol immersion lenses with correction collars are used (RI closer to that of the mounting media). The difference in image quality is significant. See your results point 3.4.2! I would appreciate seeing the performance of the machines under less aberrant conditions.
    Please comment on that.
  1. Line 169: It is not clear whether the authors used the SP5 or the SP8 for comparison of confocal systems. Anyway, which detector system was used? There is a big difference between the performance of a standard PMT and a hybrid detector concerning image noise and the required setup (e.g. less laser intensity required). Which detector system has been used for the point scanner system?
  1. Line 194: Closing the pinhole to Airy 0.49 will significantly decrease the number of detected photons and, thus, decrease the SNR relevant for post deconvolution. There is no need to use setup (makes sense when imaging very thin objects). One can also use Airy 1 and deconvolve it. Optimized deconvolution per se can increase the resolution of image objects. Please note: the Schrader et al. paper was done with gold-particles. The results can hardly be compared with fluorescently labeled samples.
  1. Line 196, line 293: Deconvolution was performed with a theoretical PSF. However, especially under the imaging condition used in this work (high RI differences), a measured PSF is required. The authors may measure a bead (which is also challenging if performed correctly) and compare the diffraction pattern with a theoretical PSF model generated with the metadata of the data sets (e.g. using the Huygens Nyquist calculator). The differences are typically significant. Although using the same objective, the PSF of the machines can differ significantly.
    Please comment on that.
  1. Line 222: What is the criteria/threshold for 'minimizing' bleaching in the different microscopy techniques (1%, 5 %, 10% bleach rate ?)?

  1. Line 249: The dwell-time also affects photon gathering. What was the dwell time in the Leica setup? Why did the authors not use e.g. line accumulation and post deconvolution for the generation of the images?
  1. Line 255: As far I know HyVolution is not more purchased by Leica.
    Please clarify this in the text.
  1. Line 257: [35-37]. These numbers point to the wrong references in the reference list.
  2. Line 291: Which Leica? SP5 or SP8?
  1. Line 293-294: What is the rationale for using CLME? There are other algorithms available in the Huygens package which may be useful. This algorithm is typically used for noisy image data; according to the text, the images show high SNR. Why 40 and not e.g. 30 iterations? The number of iterations affects the result.
    Please comment on that.
  1. Line 295-296: To what extent were the SNR parameters changed? The SNR setting typically significantly affects the deconvolution result. Besides, the SNR has first to be calculated (there are different methods of how to do this including the automated way provided by Huygens) and then the setting is adapted to this value.
  2. Figure 1B: Careful inspection of this image reveals that the two proteins do not co-localize completely. There is a clear unidirectional shift of the red channel to the left observable, most likely due to a hardware-based misalignment.
    Please comment on that?
  3. Line 328-329: This statement is confusing. The Nyquist criterion defines the minimum sampling frequency required for reliable reconstruction of the measured analogous signal (~2.3 smaller than the lowest frequency in the signal = lateral resolution limit); the resolution limit is related to the wavelength and numerical aperture.
    Please clarify this in the text.

  1. Table1: The computation time for the data sets depends strongly on the hardware of the individual machines. This is also true for CUDA computing. However, several companies offer different hardware configurations of their workstations. The distinct hardware configurations are not indicated. Furthermore, acquisition time depends e.g. on the set physical speed in case of confocal systems, line averaging, etc.; also the speed of the stage should not be underestimated in case of mosaic imaging mode). Information on the specific hardware configurations and acquisition (speed) settings required for any comparative analysis are missing; the possibility of confocal high-speed resonant scanning was ignored.
  1. Figure 3-A1-C1: From the images, the reader has the impression that the axial detection capability of the Zeiss Apoptome and Zeiss Spinning Disk is superior to those of the Leica confocal (not discussed in the text). However, according to the text the pinhole was set to Airy 0.5. Decreasing the physical pinhole size will improve the axial resolution of the microscope to some extend but on the cost of light detection. The number of detected photons is significantly decreased under this condition. Setting the pinhole to Airy 1 would significantly increase the brightness of the image axially and would presumably lead to similar results than shown in Fig. 3A1-B1. This little change has a great effect on the image quality. Acquisition speed settings are not shown (what about confocal resonant scanning?). There are no particles ‘resolved’(just one patch visible). Thus, I think one cannot compare the power of the machine seriously in this way.
    Please comment on that.

  2. Line 651-653: Computing of small datasets on GPUs is inefficient as the data transfer overhead can be significant; this is not a new finding but published elsewhere. Please search for ‘GPU transfer overhead’ in the literature and cite relevant works.

  3. Line 692: ‘STED microscopy imaging definitely delivered the best optical resolution (Fig. 4C)’. There was no structure resolved in this figure (i.e. separation of two closely associated image objects). Furthermore, the scale bar in the STED image differs from those shown in A and B, presumably because the 100x objective was used for STED; since the size of the particles is similar than in the other images, does this mean that using STED one can detect smaller particles?
    Please comment on that.
  1. Line 701-703: ‘Our study clearly revealed that deconvolution is a powerful tool to improve image quality and…’. This is not a new finding and already published elsewhere. Please search for ‘deconvolution’ and in the context of the specific microscopic technique in the literature and cite them.

  2. Line 704: However, line averaging can be accompanied by a loss of information (smooth-out of structures).
  1. Line 705-707: ‘averaging was not necessary…’. Deconvolution (e.g. CMLE) does not only aim at compensating the PSF but removes efficiently image noise. Thus, typically no averaging is necessary, also not in confocal mode (see the Huygens manual and cite it).

Round 2

Reviewer 1 Report

I thank the authors for correcting the oversights and responding to the reviewer's comments. I understand that it is hard to compare these different techniques in a quantitative manner, and I do believe the new supplemental figures are a great contribution. Especially Fig S2 is deserving to be a main figure in my opinion. While most of the issues mentioned have been addressed, I do still have a few comments on the current version.

  • An introduction is meant to set the stage for the rest of the paper, to frame the context of the current work. For a special issue, this long introduction about ageing may have made sense, but as it is not the case anymore, I do think the current version raises some false expectations of what the topic of this paper will be. As also Reviewer 3 pointed out, imaging of brain specimens is challenging for a variety of reasons, and this paper is relevant to a much broader readership than aging researchers. Therefore, I do believe the manuscript would strongly benefit from an updated introduction.
  • I thank the authors for revising figure 3, however, the legend has not been updated to fit the new figure. As it appears on my screen, not a whole lot can be seen in the panels A4, B4 and C3. Are the optimal brightness/contrast settings used here?

Reviewer 3 Report

The authors now substantially changed their strategy on how to commit the content of their work. Recommendations concerning critical aspects in the text were accepted, and questions sufficiently answered. Furthermore, there is new data provided. Relevant metadata/detailed microscope settings are available on request. Obligatory affiliations were correctly assigned. 

Importantly, the authors have added a paragraph in the discussion section, which critically addresses the limitations in the comparison of such microscope systems. It has been clarified that the shown results do not necessarily represent the ultimate quality, which can be achieved with the individual systems. Consequently, readers can realize that the evaluation of such machines has a somewhat subjective character. Nevertheless, scientists who are non-experts in the imaging field get an overview of a selection of optical imaging techniques and their capacities for brain studies.

I have no further questions.
